# Clinical Effectiveness of Mirogabalin Besylate for Trigeminal Neuropathy after Skull Base Surgery: Illustrative Cases

**DOI:** 10.3390/medicines10080048

**Published:** 2023-08-17

**Authors:** Kosuke Karatsu, Ryota Tamura, Tsubasa Miyauchi, Junki Sogano, Utaro Hino, Takashi Iwama, Masahiro Toda

**Affiliations:** Department of Neurosurgery, Keio University School of Medicine, 35 Shinanomachi, Shinjuku-ku, Tokyo 160-8582, Japan; chilipep1221@gmail.com (K.K.); tsubasa-prokofiev@keio.jp (T.M.); jnk.sogano@gmail.com (J.S.); utaro0526@keio.jp (U.H.); tks.iwama416@gmail.com (T.I.); todam@keio.jp (M.T.)

**Keywords:** mirogabalin besylate, neuropathy, skull base, schwannoma, meningioma, postoperative

## Abstract

Background: Postoperative trigeminal neuropathy may be seen after surgery for middle and posterior cranial fossa lesions. Although neuropathic pain is a cause of reduced quality of life, global consensus on postoperative pain management is lacking. Mirogabalin besylate is a selective ligand for the α2δ subunit of voltage-gated calcium channels. Although mirogabalin has been used for patients with postherpetic neuralgia and painful diabetic peripheral neuropathy, few reports have assessed the effect on postsurgical neuropathy. In this report, we describe a clinical effectiveness of mirogabalin for trigeminal neuropathy after skull base surgery. Case description: Case 1: A 51-year-old female with right trigeminal schwannoma was operated on via the anterior transpetrosal approach. She had tingling and numb feelings in the right face postoperatively. Mirogabalin was orally administered after the operation. Her continuous facial numbness immediately improved. Case 2: A 55-year-old female with left middle fossa base meningioma extending into the infratemporal fossa was operated on via the infratemporal fossa approach. She had a tingling feeling in the left face postoperatively. Mirogabalin was orally administered for this symptom after the operation, which gradually improved. Conclusions: Mirogabalin may show significant pain relief for patients with trigeminal neuropathy after skull base surgery. Further studies using a larger number of patients are warranted to confirm these findings.

## 1. Introduction

Skull base surgery is a highly specialized field that addresses tumors and other abnormalities on the base of the skull. Surgery at the skull base is frequently challenging due to its constrained surgical field and the proximity of crucial neural and vascular structures. The manipulation of the skull base entails the inherent risk of temporary or permanent neurologic deficits [1]. The incidence of iatrogenic cranial nerve injuries in skull base surgeries across various surgical techniques ranges from 2% to 47% [2,3]. Neuropathic pain is difficult to manage because of the heterogeneity of its etiology, symptoms, and underlying mechanisms. Patients with neuropathic pain may experience shooting, burning pain, which may be constant or intermittently occur. A feeling of tingling, numbness, or a loss of sensation are also common for patients. Trigeminal neuralgia, postherpetic neuralgia, and diabetic neuropathy are the most common causes of peripheral neuropathic pain. In general, neuropathic pain following trigeminal nerve injury is a chronic pain condition that is the most problematic consequence of dental or oromaxillofacial surgical procedures with major medico-legal implications. Postoperative trigeminal neuropathy may be also seen after surgery for middle and posterior cranial fossa lesions. For general trigeminal neuropathy, carbamazepine and oxcarbazepine are often considered as the first-line choices. Other options include anticonvulsants like lamotrigine and gabapentin, muscle relaxants such as type A botulinum toxin and baclofen, and further consideration may be given to pregabalin. If pain remains unrelieved despite pharmacotherapy, surgical intervention should be contemplated, with microvascular decompression of the trigeminal nerve being the most typical surgical approach [4,5].

The anterior transpetrosal approach is employed when dealing with lesions in the cerebellopontine angle or anterior pontine region. It is particularly well-suited for trigeminal schwannomas or tentorial meningiomas extending into Meckel’s cave. The infratemporal fossa approach is a method of accessing lesions located just beneath the middle cranial fossa from a lateral direction. The infratemporal fossa is located at the boundary between the neck and the cranial base, involving multiple medical specialties such as neurosurgery, otolaryngology, and maxillofacial surgery. The mandibular branch of the trigeminal nerve passes through the oval foramen, entering this region.

Both skull base approaches are sometimes associated with trigeminal nerve neuropathy. Painful trigeminal neuropathy often manifests as a prolonged or nearly continuous condition. Commonly, it is described as a sensation of scorching heat, constriction, or a pain akin to being punctured by needles. Painful trigeminal neuropathy is difficult to treat pharmacologically [6]. Although neuropathic pain is a cause of reduced quality of life, global consensus on postoperative pain management is lacking [6].

Mirogabalin besylate (mirogabalin, Tarlige^®^), developed by Daiichi Sankyo, represents a gabapentinoid formulation approved in Japan. Previous studies have shown that α2δ subunits of voltage-gated calcium channels are significant for the trafficking and physiological function of the calcium channels. α2δ-1 subunits of voltage-gated calcium channels play a key role during neuropathic pain progression, which results from the injury to sensory nerves. α2δ-1 subunits of voltage-gated calcium channels are strongly up-regulated in somatosensory neurons following nerve damage. α2δ-1 and α2δ-2 subunits were identified as the therapeutic targets for the gabapentinoids [7]. Mirogabalin selectively binds to the α2δ subunit of voltage-gated calcium channels at the presynaptic terminal [7]. In comparison to pregabalin, mirogabalin has been shown to possess higher selective affinity for the α2δ-1 subunit, along with a slower dissociation rate, resulting in a potent and enduring analgesic effect. Additionally, due to its rapid dissociation from the α2δ-2 subunit, which is potentially implicated in central nervous system-specific adverse events, mirogabalin exhibits a favorable profile from the perspective of side effect risk. The therapeutic application of mirogabalin is commonly employed in prevalent conditions characterized by intense pain, such as postherpetic neuralgia and painful diabetic peripheral neuropathy [8,9]. However, few reports have assessed the effect of mirogabalin in patients with postsurgical neuropathy. The key treatment goals for painful neuropathy are to reduce pain, maintain or improve function, and preserve or improve quality of life. Although alleviation of pain is often difficult to achieve, pharmacotherapy is the foundation of neuropathy therapy. In this report, we first describe the clinical effectiveness of mirogabalin for trigeminal sensory neuropathy after skull base surgery.

## 2. Case Description

### 2.1. Case 1

A 51-year-old female presented with right facial hypoesthesia in V2 and V3 sensory areas. Right petroclival extra-axial contrast-enhancing lesion exhibited mildly heterogenous (T1 hypointense/T2 hyperintense) signal intensities with small cystic areas (Figure 1A). Fast imaging employing steady-state acquisition (FIESTA) demonstrated tumor extension into Meckel’s cave (Figure 1A). Trigeminal schwannoma was suspected. Anterior transpetrosal approach was performed for tumor removal [10] (Figure 1B). The epidural detachment of the middle cranial fossa was performed towards the petrous ridge. The middle meningeal artery within the foramen spinosum was coagulated and dissected. The dura mater directly above the greater petrosal nerve was interdurally detached, preserving the nerve. The periosteal dura surrounding the trigeminal nerve was sharply incised, and further elevation of the middle cranial fossa dura was achieved. Upon sufficient epidural dissection beyond the arcuate eminence, the petrous ridge was reached, exposing Kawase’s quadrangle, a landmark for drilling the petrous apex. The petrous bone apex was removed to access the posterior cranial fossa. The temporal dura and posterior cranial fossa dura were incised parallel to the superior petrosal sinus, respectively, followed by coagulation and detachment of the superior petrosal sinus. The tentorium was incised while preserving the trochlear nerve, exposing the trigeminal nerve on the surface of the petroclival tumor component. In this process, the porus trigeminus was released to expose the trigeminal ganglion within Meckel’s cave. The tumor was removed subcapsularly while preserving the trigeminal nerve (Figure 1A,C). The pathological diagnosis revealed schwannoma.

The patient experienced mild tingling and numb feelings in the right face postoperatively. Although oral non-steroidal anti-inflammatory drugs (NSAIDs) and/or acetaminophen were administered for five days, they were not effective in reducing her symptoms. Oral formulations of mirogabalin are available in 2.5, 5, 10, and 15 mg tablets with a recommended starting dose of 5 mg twice daily with weekly increases by 5 mg up to a maximum dose of 15 mg. Therefore, mirogabalin (total 10 mg, 5 mg twice daily) was orally administered six days after the operation. Her continuous tingling and numb feelings improved (verbal rating scale [VRS]: 2 to 0 [11], modified Rankin Scale [mRS]: 1 to 0 [12]). The VRS is a four-point scale and consists of a list of adjectives describing various levels of pain intensity (0 = no pain, 1 = mild pain, 2 = moderate pain, and 3 = severe pain) [11,13]. She was discharged from the hospital fourteen days after the operation and stopped taking mirogabalin on her own judgement without talking to the doctor. Because her symptom recurred, she started taking mirogabalin again as an outpatient 33 days after the operation. Her symptom improved, and no adverse effects of mirogabalin have been observed at present (130 days after the operation).

### 2.2. Case 2

A 55-year-old female presented with left facial numbness in the V3 sensory area, exhibiting a vividly enhanced lesion on the middle cranial base with homogenous (T1 hypointense/T2 isointense) signal intensities (Figure 2A). Edematous change was observed in the left temporal lobe. Middle fossa base meningioma extending into the infratemporal fossa was suspected. Preoperative axial computed tomography scan showed the tumor invasion and thickness of the middle fossa base (Figure 2B). Infratemporal fossa approach was performed for tumor removal [14]. The skin incision was curvilinear, starting at the frontal scalp and extending anterior to the external auditory canal. The zygomatic arch was resected. A frontotemporal craniotomy was performed. Bone removal extended anteriorly to include the floor of the middle fossa with tumor invasion. During the procedures, the floor of the middle cranial fossa with tumor invasion was drilled around the foramen rotundum and ovale to expose the V3 entering into the infratemporal fossa (Figure 2B,C). The lateral dural wall of the cavernous sinus was exposed. Tumor removal of the subdural tumor component was mainly performed. The pathological diagnosis of meningioma was made. Postoperative magnetic resonance imaging showed the improvement of temporal lobe edema.

The patient had a tingling feeling in the left face postoperatively. Although oral NSAIDs and/or acetaminophen were administered for two days postoperatively, they were not effective in reducing her symptoms. Therefore, mirogabalin (total 10 mg, 5 mg twice daily) was orally administered three days after the operation for the symptom, which gradually improved (VRS: 3 to 1 [11], mRS: 1 to 0 [12]). She was discharged from the hospital fourteen days after the operation, and stopped taking mirogabalin based on patient preferences. No adverse effects of mirogabalin were observed. Although her symptom recurred, she started taking an antimigraine drug for her past history as an outpatient 36 days after the operation. A slight tingling feeling persists at present (176 days after the operation).

For both patients, the most frequent adverse drug reactions, including nasopharyngitis, somnolence, dizziness, peripheral edema, drowsiness, and cerebellar ataxia [7], were not observed.

## 3. Discussion

Neuropathic pain is a problem that plagues many patients. The efficacy and safety of mirogabalin have been previously demonstrated for several types of neuropathic pain, such as diabetic peripheral neuropathic pain [8] and postherpetic neuralgia, which are common clinical conditions with intense distress for patients. [9]. In both studies, mirogabalin showed significant pain relief, with a greater decrease in pain score as the daily dose of mirogabalin increased. All mirogabalin doses tested were well tolerated. Recent case reports have demonstrated that mirogabalin is effective for patients with chronic trigeminal neuralgia, such as trigeminal trophic syndrome [15,16,17,18,19]. Mirogabalin may be efficiently used for patients with general trigeminal neuralgia in the chronic stage. However, there are few reports associated with postsurgical neuropathy. 

In general, the clinical guideline recommends NSAIDs, gabapentin or pregabalin, and ketamine for post-operative pain management [20]. First, acetaminophen and NSAIDs are recommended in patients without contraindications, which inhibit prostaglandin synthesis by inhibiting cyclooxygenase-1. Celecoxib is a cyclooxygenase-2 inhibitor and reduces pain by preventing inflammation, causing hyperalgesia and allodynia. The most frequent side effects of NSAIDs are gastrointestinal symptoms. Celecoxib prevents gastrointestinal ulcer complications compared with other NSAIDs. The principal mechanism of action of carbamazepine is the blockade of inactivated neuronal sodium channels, preventing them from opening. The side effects of carbamazepine include feeling sleepy, feeling dizzy, headaches, and feeling or being sick. Gabapentin binds to the α² subunit of voltage-dependent Ca2+ channels, and prevents the development of central excitability by stabilizing the neuronal membrane and decreasing the subcutaneous response to signals of pain fiber. Side effects of gabapentin are somnolence, dizziness, confusion, and ataxia. The primary mechanism of ketamine is N-methyl-D-aspartate receptor antagonism. Ketamine exerts preventative analgesic effects by modulating central sensitization and decreasing post-operative pain. Ketamine use can include flashbacks, memory loss, and problems with concentration. Pregabalin is an antagonist of voltage-gated Ca2+ channels and specifically binds to the alpha-2-delta subunit to produce antiepileptic and analgesic actions. Pregabalin may cause blurred vision, double vision, clumsiness, unsteadiness, dizziness, drowsiness, or trouble with thinking [20].

Recently, mirogabalin was assessed in patients with thoracic postsurgical pain [21]. Oral anti-inflammatory analgesics can also be administered postoperatively and have been shown to be beneficial in postoperative surgical patients without contraindications. For severe postoperative pain, treatments vary by region, and consensus on a global management protocol is lacking. In this report, mirogabalin has been shown to be beneficial in postneurosurgical patients with trigeminal sensory neuropathy. Mirogabalin treatment was superior to conventional anti-inflammatory analgesics (NSAIDs and/or acetaminophen) pharmacotherapy.

In the area of skull base surgery, painful trigeminal neuropathy is known to be difficult to treat pharmacologically [6]. Neuropathic pain may contribute to disability, depression, anxiety, and sleep disorders. Therefore, neuropathic pain is a global cause of reduced quality of life. In addition to carbamazepine, gabapentin and pregabalin have become the standard drugs in treating trigeminal neuropathy [22,23]. However, conventional gabapentinoids, gabapentin and pregabalin, bind to the α2δ-1 and α2δ-2 subunits nonselectively, and produce adverse drug reactions, such as dizziness, ataxia, somnolence, and headache [22,23]. Mirogabalin exhibits a heightened affinity for the α2δ-1 subunit and a slower dissociation, thereby yielding a more prolonged analgesic effect. Conversely, its affinity for the α2δ-2 subunit is lower, and dissociation occurs more rapidly, resulting in a diminished risk of central nervous system-specific adverse effects (Figure 3) [24,25,26]. Furthermore, recent investigations have demonstrated favorable tolerability upon transitioning from pregabalin to mirogabalin, endorsing its efficacy in the treatment of peripheral neuropathic pain [27]. The majority of patients could be incrementally escalated to therapeutic doses. Within this study, a significant reduction in the overall mean Visual Analogue Scale scores was observed [27]. In our two cases, mirogabalin had a balanced efficacy, and may provide an alternative therapeutic option for the treatment of trigeminal sensory neuropathy after skull base surgery. The most frequent adverse drug reactions, including nasopharyngitis, somnolence, dizziness, peripheral edema, drowsiness, and cerebellar ataxia, were not observed. However, careful dosing and titration may be necessary. In particular, dizziness and somnolence are side effects which may occur in a certain proportion, which is why it is recommended to refrain from taking the medicine before driving. For individuals leading a daily life that necessitates driving, there is a potential challenge in timing administration to achieve the desired analgesic effect. Furthermore, in the case of elderly individuals, it demands attention due to the increased risk of falls. Additionally, the medication is comparatively more expensive than typical anti-inflammatory analgesics, posing an economic burden [28].

Neuropathic pain represents not a singular ailment, but rather a syndrome induced by a multitude of diverse pathogenic mechanisms. Recently, integrated multimodal treatment with the current treatment facility, including various medical disciplines, were recommended for chronic neuropathic pain based on the patient’s medical history. Further studies are needed to establish an integrated, cause-specific, cost-effective approach for patients after skull base surgery [29].

## 4. Limitation

Herein, we report the usage of mirogabalin in patients that suffer mild trigeminal neuropathy. As both patients were aware of taking medication for neuropathic pain, there was a possibility of the placebo effect intervening. Therefore, we must discuss confounding factors that might have influenced the observed outcomes. Mirogabalin is expected to be an important new treatment option for the patients with trigeminal neuropathy after skull base surgery, which should be followed by a study that includes patients with severe neuropathy. Admittingly, it may be that mirogabalin provided some/minor relief to these two patients to help to accelerate the course. In general, NSAIDs and acetaminophen are effective for wound pain but not for neuropathic pain. Therefore, the effect of mirogabalin should be compared with other antineuropathic drugs, including pregabalin. In particular, pregabalin is a common prescription drug which is typically used to treat neuropathic pain, anxiety, and restless leg syndrome [30]. Detailed information (more strict follow up, formal scales of pain/life quality, etc.) is needed to more clearly understand the effect of mirogabalin. Further studies are required to determine appropriate dose adjustments on patient weight and body surface area. Furthermore, most of the studies were conducted with only Asian patients, prompting the question of whether the results are generalizable to the general population. Long-term evaluations of safety and efficacy in other races are still needed. The level of evidence of this case report is poor. The VRS is a simple tool limited to a few statements, and it appears to be the most usable tool for pain assessment in cognitively impaired subjects. However, other pain rating scales, including the Visual Analogue Scale and the Numerical/Numeric Rating Scale, are needed to evaluate the detailed clinical course [31]. A prospective study with a large number of patients is needed to confirm the efficacy and safety of mirogabalin to treat trigeminal neuropathy after skull base surgery.

## 5. Conclusions

Mirogabalin may show significant pain relief for patients with trigeminal neuropathy after skull base surgery. Mirogabalin may be expected to be an important new treatment option for patients after skull base surgery. Furthermore, it may be also helpful for not only postoperative treatment but also preoperative management for tumors involving the trigeminal nerve, such as schwannoma and meningioma. However, drug efficacy studies should be carried out by comparison with the effect of placebos or other painkillers. Further studies using a larger number of patients are warranted to confirm these findings.

## Figures and Tables

**Figure 1 medicines-10-00048-f001:**
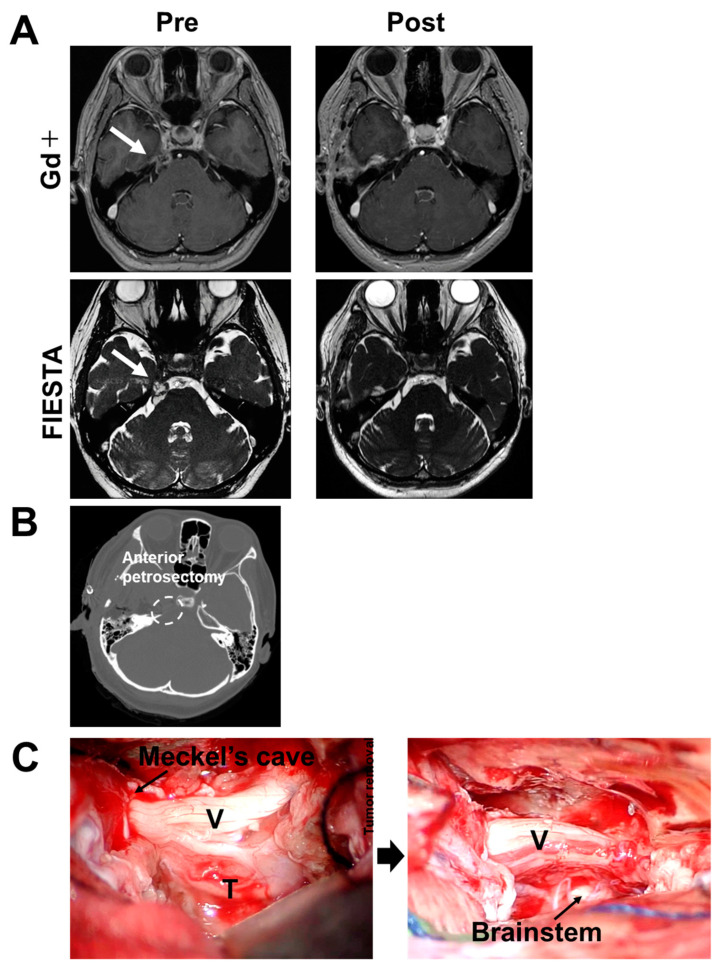
Clinical characteristics in Case 1. (**A**) Pre- and postoperative radiographic images are shown. T1-weighted axial MRI with gadolinium enhancement reveals a large, heterogeneously enhancing, dumbbell-shaped lesion. FIESTA demonstrates an extra-axial mass at the right petroclival region, extending to Meckel’s cave. Gd, gadolinium; FIESTA, fast imaging employing steady-state acquisition. White arrow: tumor. (**B**) Postoperative CT bone window images show the area of anterior petrosectomy. (**C**) Intraoperative findings of the ATP are shown. The tentorium was opened, exposing the trigeminal nerve including the tumor (**left** panel). Subcapsular removal of the tumor and decompression of Meckel’s cave were carried out (**right** panel). T, tumor; V, trigeminal nerve.

**Figure 2 medicines-10-00048-f002:**
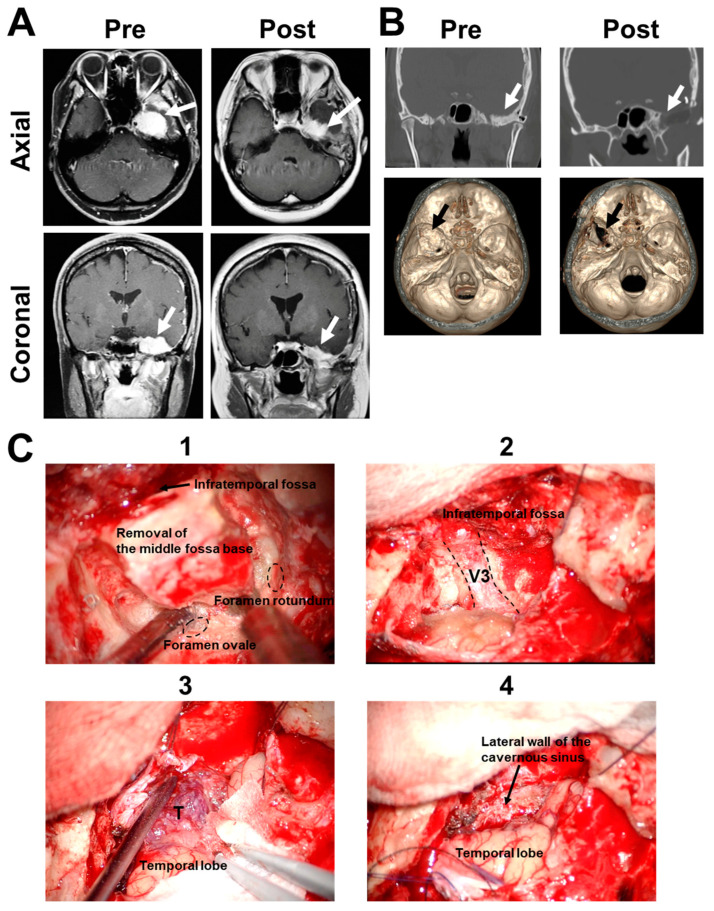
Clinical characteristics in Case 2. (**A**) Pre- and postoperative radiographic images are shown. T1-weighted axial and coronal MRI with gadolinium enhancement reveals an extra-axial mass located in the left middle cranial fossa involving the left cavernous sinus and infratemporal fossa. White arrow, tumor. (**B**) Pre-and postoperative CT bone window images (upper panel, corona view; lower panel, three-dimensional) show the drilling area of the middle fossa base (white and black arrows). (**C**) Intraoperative findings of the infratemporal fossa approach are shown. The craniotomy was created along the middle fossa base to expose the foramen rotundum and ovale epidurally. The middle fossa base was drilled off until the penetrating tumor was exposed (1). Final view of the infratemporal fossa approach showing the contents around the foramen ovale (2). Tumor removal of the subdural tumor component was achieved (3). The lateral dural wall of the cavernous sinus was exposed (4). T, tumor; V3, mandibular nerve.

**Figure 3 medicines-10-00048-f003:**
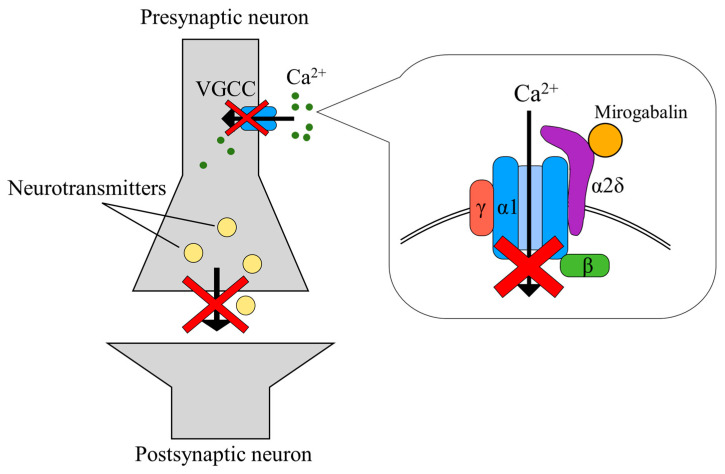
Mechanism of action of mirogabalin. Mirogabalin is a specific ligand that binds to the α2δ subunit of voltage-gated Ca channels, particularly targeting the α2δ-1 subunit, which plays a crucial role in neuropathic pain. By binding to the α2δ subunit and inhibiting the influx of Ca ions, it reduces the excessive release of excitatory neurotransmitters, thus expressing its analgesic effects.

## Data Availability

The original contributions presented in the study are included in the article; further inquiries can be directed to the corresponding author.

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
