# Peer review of "Clinical Effectiveness of Mirogabalin Besylate for Trigeminal Neuropathy after Skull Base Surgery: Illustrative Cases"

_medicines, 2023, doi:10.3390/medicines10080048_

Round 1
Reviewer 1 Report
The present manuscript reports about two skull base surgery cases who suffer from postoperative trigeminal neuropathy. The authors used mirogabalin, which is a novel gabapentinoid with approval in Japan.
Introduction: The authors adequately describe the incidence of iatrogenic cranial nerve injuries and it´s implication on quality of life. Furthermore, they give the readership insight into mirogabalin.
Cases: Cases are precisely described regarding preoperative images, intraoperative technique, and postoperative course
Discussion: The authors debate the standardized and routinely used drugs in the clinical care for skull base surgery patients.
Language: The spelling and writing is fine.
All in all, the authors describe a new potential drug. However, the level of evidence of this study is poor and it is only a case report. There won´t be an impact of the present study for the clinical care of skull base surgery patients.
Author Response
We are very grateful to the reviewers for their insightful comments and suggestions, which would undoubtedly help us to improve our manuscript immensely. As indicated in the responses below, we have taken all their comments and suggestions into account when generating the revised version of the manuscript. Responses to the reviewers’ comments appear after the arrows, in blue text.
Reviewer 1:
The present manuscript reports about two skull base surgery cases who suffer from postoperative trigeminal neuropathy. The authors used mirogabalin, which is a novel gabapentinoid with approval in Japan.
→
Thank you very much for your review.
- Introduction: The authors adequately describe the incidence of iatrogenic cranial nerve injuries and it´s implication on quality of life. Furthermore, they give the readership insight into mirogabalin.
→
Thank you very much for your review. As the reviewer indicated, we have described the relationship between iatrogenic cranial nerve injury and quality of life. In addition, we have added the explanation about mirogabalin besylate.
- Cases: Cases are precisely described regarding preoperative images, intraoperative technique, and postoperative course
→
Thank you very much for your review. As the reviewer indicated, we have added detailed explanation about postoperative course using functional status scale.
- Discussion: The authors debate the standardized and routinely used drugs in the clinical care for skull base surgery patients.
→
Thank you very much for your review. As the reviewer indicated, we have added more explanation about routinely used drugs for skull base surgery patients.
- Language: The spelling and writing is fine.
→
Thank you very much for your comments.
- All in all, the authors describe a new potential drug. However, the level of evidence of this study is poor and it is only a case report. There won´t be an impact of the present study for the clinical care of skull base surgery patients.
→
Thank you very much for your review. Tumors located on the skull base are relatively rare but can be devastating to patients. Therefore, we think that case reports will be helpful for the skull base surgeons. However, as the reviewer indicated, the level of evidence of this case report is poor. Therefore, we have added the limitation section.
Limitation
Herein we report the usage of mirogabalin in patients that suffer mild trigeminal neuropathy. As both patients are aware of taking medication for neuropathic pain, there is a possibility of the placebo effect intervening. Therefore, we must discuss confounding factors that might have influenced the observed outcomes. Mirogabalin is expected to be an important new treatment option for the patients with trigeminal neuropathy after skull base surgery, which should be followed by a study that includes patients with se-vere neuropathy. Admittingly, it might only be that mirogabalin allowed some/minor relief to these two patients to give some help to accelerate the course. In general, NSAIDs and acetaminophen are effective for wound pain but not for neuropathic pain. Therefore, the effect of mirogabalin should be compared with other antineuropathic drugs in-cluding pregabalin. In particular, pregabalin is a common prescription drug, which is typically used to treat neuropathic pain, anxiety, and restless leg syndrome [18]. Detailed information (more strict follow up, formal scales of pain/life quality, etc.) is needed to more clearly understand the effect of mirogabalin. Further studies are required to de-termine appropriate dose adjustments on patient weight and body surface area. Fur-thermore, most of the studies were conducted with only Asian patients, prompting the question if the results are generalizable to the general population. Long-term evalua-tions of safety and efficacy in other races are still needed. The level of evidence of this case report is poor. The VRS is a simple tool limited to a few statements, and it appears to be the most usable tool for pain assessment in cognitively impaired subjects. However, other pain rating scales, including Visual Analogue Scale and Numerical/numeric Rating Scale are needed to evaluate detailed clinical course [Williamson A. J Clin Nurs. 2005]. A prospective study with large number of patients is needed to confirm the efficacy and safety of mirogabalin to treat trigeminal neuropathy after skull base surgery.
Williamson A, Hoggart B. Pain: a review of three commonly used pain rating scales. J Clin Nurs. 2005:14:798-804.

Reviewer 2 Report
Dear Authors,
Congratulations on the manuscript. The manuscript is really interesting and has been structured very nicely. I don't have many more comments but I would like to add a few points.
-I will recommend mentioning a few other post-operative medications in the discussion part and comparing the efficacy and safety of those drugs in brief.
-Can you demonstrate schematically the action of the drugs in neurons using the units if possible?
-Mention the challenges the patient will face using neuropathic drugs,like add some economic issue etc
Please follow this article to know more about the recent challenges about neuropathic pain
Hange, N., Poudel, S., Ozair, S., Paul, T., Nambakkam, M., Shrestha, R., ... & Patel, P. (2022). Managing Chronic Neuropathic Pain: Recent Advances and New Challenges. Neurology Research International, 2022.
Author Response
We are very grateful to the reviewers for their insightful comments and suggestions, which would undoubtedly help us to improve our manuscript immensely. As indicated in the responses below, we have taken all their comments and suggestions into account when generating the revised version of the manuscript. Responses to the reviewers’ comments appear after the arrows, in blue text.
Reviewer 2:
Congratulations on the manuscript. The manuscript is really interesting and has been structured very nicely. I don't have many more comments but I would like to add a few points.
→
Thank you very much for your review.
1.I will recommend mentioning a few other post-operative medications in the discussion part and comparing the efficacy and safety of those drugs in brief.
→
Thank you very much for your review. We have added the detailed information of other post-operative medications in the revised manuscript.
In general, the clinical guideline recommends nonsteroidal anti-inflammatory drugs (NSAIDs), gabapentin or pregabalin, and ketamine in the post-operative pain management [Vadivelu N.J Pain Res. 2016]. First, acetaminophen and NSAIDs are recommended in patients without contraindications, which inhibit prostaglandin synthesis by inhibiting cyclooxygenase-1. Celecoxib is a cyclooxygenase-2 inhibitor and reduces pain by preventing inflammation, causing hyperalgesia and allodynia. The most frequent side effects of NSAIDs are gastrointestinal symptoms. Celecoxib prevents gastrointestinal ulcer complications compared with other NSAIDs. The principal mechanism of action of carbamazepine is the blockade of inactivated neuronal sodium channels, preventing them from opening. The side effects of carbamazepine include feeling sleepy, feeling dizzy, headaches and feeling or being sick. Gabapentin binds to the α² subunit of voltage-dependent Ca2+ channels, and prevents the development of central excitability by stabilizing the neuronal membrane and decreasing the subcutaneous response to signals of pain fiber. Side effects of gabapentin are somnolence, dizziness, confusion, and ataxia. The primary mechanism of ketamine is N-methyl-D-aspartate receptor antagonism. Ketamine exerts preventative analgesic effects by modulating central sensitization and decreasing post-operative pain. The ketamine use can include flashbacks, memory loss and problems with concentration. Pregabalin is an antagonist of voltage gated Ca2+ channels and specifically binds to alpha-2-delta subunit to produce antiepileptic and analgesic actions. Pregabalin may cause blurred vision, double vision, clumsiness, unsteadiness, dizziness, drowsiness, or trouble with thinking.
Vadivelu N, Kai AM, Tran D, Kodumudi G, Legler A, Ayrian E. Options for perioperative pain management in neurosurgery.J Pain Res. 2016;9:37-47.
2.Can you demonstrate schematically the action of the drugs in neurons using the units if possible?
→
Thank you very much for your review.
As the reviewer indicated, we have demonstrated schematically the action of the drugs and cited the appropriate reference.
Figure legend:
Mirogabalin is a specific ligand that binds to the α2δ subunit of voltage-gated calcium channels, particularly targeting the α2δ-1 subunit, which plays a crucial role in neuropathic pain. By binding to the α2δ subunit and inhibiting the influx of calcium ions, it reduces the excessive release of excitatory neurotransmitters, thus expressing its analgesic effects.
ZajÄ…czkowska R, Mika J, Leppert W, Kocot-KÄ™pska M, Malec-Milewska M, Wordliczek J. Mirogabalin-A Novel Selective Ligand for the α2δ Calcium Channel Subunit. Pharmaceuticals (Basel). 2021;14:112.
3.Mention the challenges the patient will face using neuropathic drugs, like add some economic issue etc
→
Thank you very much for your comments. We have described the economic issue of the neuropathic drugs in the revised manuscript.
Dizziness and somnolence are known side effects, occurring in a certain proportion, which is why it is recommended to refrain from taking the medicine before driving. For individuals leading a daily life that necessitates driving, there is a potential challenge in timing the administration to achieve the desired analgesic effect. Furthermore, in the case of elderly individuals, it demands attention due to the increased risk of falls. Additionally, the medication is comparatively more expensive than typical anti-inflammatory analgesics, posing an economic burden [Finnerup NB, Lancet Neurol. 2015].
Finnerup NB, Attal N, Haroutounian S, McNicol E, Baron R, Dworkin RH, Gilron I, Haanpää M, Hansson P, Jensen TS, Kamerman PR, Lund K, Moore A, Raja SN, Rice AS, Rowbotham M, Sena E, Siddall P, Smith BH, Wallace M. Pharmacotherapy for neuropathic pain in adults: A systematic review and meta-analysis and updated NeuPSIG recommendations. Lancet Neurol 2015;142:162–73.
4.Please follow this article to know more about the recent challenges about neuropathic pain
Hange, N., Poudel, S., Ozair, S., Paul, T., Nambakkam, M., Shrestha, R., ... & Patel, P. (2022). Managing Chronic Neuropathic Pain: Recent Advances and New Challenges. Neurology Research International, 2022.
→
Thank you very much for your comments. According to the reviewer’s comment, we have cited the indicated reference in the revised manuscript.
Neuropathic pain is not a single disease, but instead a syndrome that can be caused by a number of diverse etiologies. Recently, integrated multimodal treatment with the current treatment facility, including various medical disciplines, were recommended for the chronic neuropathic pain based on the patient’s medical history. Further studies are needed to establish the integrated, cause-specific, cost-effective approach for the patients after skull base surgery [Hange N. Neurol Res Int. 2022].
Hange N, Poudel S, Ozair S, Paul T, Nambakkam M, Shrestha R, Greye F, Shah S, Raj Adhikari Y, Thapa S, Patel P. Managing Chronic Neuropathic Pain: Recent Advances and New Challenges. Neurol Res Int. 2022;2022:8336561.

Reviewer 3 Report
This is nicely written papers with two case reports of postsurgical trigeminal neuralgia.
Thwe surgical part is well described and documented.
But, surprisingly, the description of the neuropathic pain treatment in two idividual cases is rather laconic. First of all, the verbal rating scale for assesment of pain intensit yis not a best choice, but acceptable. In such a case, the range of th scale must be expicitely specified - there was a drop of pain intensity from 2 to 0, but was was the maximum on the scale? This is even more necessary in the 2nf case, in whom there was npot achieved elimination of pain but a reduction from 2 to 1. Usually as clinically meaningful is considered 30 or at least 20% of reduction on any scale. Another question is, why the therapy continued so long in the 1st case, even if she was free from pain? What was was the final result, functional status and quality of life at followup? Eg. modified Rankin scale results? These issues are worth to be mentioned in both cases! After this small adjustments I recommend this text to publication.
Author Response
We are very grateful to the reviewers for their insightful comments and suggestions, which would undoubtedly help us to improve our manuscript immensely. As indicated in the responses below, we have taken all their comments and suggestions into account when generating the revised version of the manuscript. Responses to the reviewers’ comments appear after the arrows, in blue text.
Reviewer 3:
This is nicely written papers with two case reports of postsurgical trigeminal neuralgia.
The surgical part is well described and documented.
But, surprisingly, the description of the neuropathic pain treatment in two idividual cases is rather laconic. First of all, the verbal rating scale for assesment of pain intensit yis not a best choice, but acceptable. In such a case, the range of th scale must be expicitely specified - there was a drop of pain intensity from 2 to 0, but was was the maximum on the scale? This is even more necessary in the 2nf case, in whom there was npot achieved elimination of pain but a reduction from 2 to 1. Usually as clinically meaningful is considered 30 or at least 20% of reduction on any scale. Another question is, why the therapy continued so long in the 1st case, even if she was free from pain? What was was the final result, functional status and quality of life at followup? Eg. modified Rankin scale results? These issues are worth to be mentioned in both cases! After this small adjustments I recommend this text to publication.
→
Thank you very much for your review. As the reviewer indicated, we have added the detailed clinical course in the revised manuscript.
In our hospital, VRS was routinely used to evaluate postsurgical pain. The VRS is a simple tool limited to a few statements, and it appears to be the most usable tool for pain assessment in cognitively impaired subjects. However, other pain rating scales, including Visual Analogue Scale and Numerical/numeric Rating Scale are needed to evaluate detailed clinical course [Williamson A. J Clin Nurs. 2005]. We have added the limitation in the revised manuscript.
In Case 2, the highest VRS score is 3 (average: 2). In Case 2, mirogabalin was orally administered three days after the operation for the symptom, which gradually improved (VRS: 3 to 1). In Case 1, she was discharged from the hospital fourteen days after the operation and stopped taking mirogabalin on her own judgement without talking to the doctor. Because her symptom recurred, she started taking mirogabalin again as an outpatient 33 days after the operation. Her symptom improved, and no adverse effects of mirogabalin were observed taking mirogabalin at present (130 days after the operation). She continues to take mirogabalin at present.
We have added the information of modified Rankin Scale [Broderick JP. Stroke. 2017] in the revised manuscript.
Williamson A, Hoggart B. Pain: a review of three commonly used pain rating scales. J Clin Nurs. 2005:14:798-804.
Broderick JP, Adeoye O, Elm J. Evolution of the Modified Rankin Scale and Its Use in Future Stroke Trials.Stroke. 2017;48:2007-2012.

Reviewer 4 Report
Summary:
The paper titled “Clinical effectiveness of mirogabalin besylate for trigeminal neuropathy after skull base surgery. Illustrative cases.” discusses the use of mirogabalin besylate, a selective ligand for the α2δ subunit of voltage-gated calcium channels, for postoperative trigeminal neuropathy after skull base surgery. The authors present two illustrative cases of patients who experienced tingling and numbness in the face after their respective surgeries, and they administered mirogabalin orally to manage the symptoms. In both cases, the patients showed improvement in their facial numbness, indicating that mirogabalin may be effective in providing pain relief for trigeminal neuropathy after skull base surgery.
Suggestions for Improvement:
Introduction : The introduction could benefit from providing a more comprehensive overview of the existing literature general pain management for trigeminal pathy including chronic conditions and inventional minimally invasive methods. Known adverse effects of mirogabalin should be stated.
Methodology: A clear description of the assessment tools used to measure pain relief would also be helpful.
Discussion: Postsurgical trigeminal neuralgia is a rare condition. What about the usage of miragabalin for chronic trigeminal neuralgia? Are there any previous studies reporting on that? The discussion section should be expanded to interpret the results and compare them with previous studies.
Please, discuss limitations of use and side effects of the drug.
It would be beneficial to discuss confounding factors that might have influenced the observed outcomes.
Conclusion: The conclusion, especially this in the abstract, is too brief; consider adding a more comprehensive summary of the study’s findings and emphasize the potential implications and significance of the results for clinical practice.
Limitations: Given that this is a case report with a limited sample size, the paper should caution readers about the generalization of the findings. This caution including a statement on the need for further research with larger cohorts should be indicated as a short "Limitations" section in the paper.
Overall, the study presents promising results regarding the effectiveness of mirogabalin besylate for trigeminal neuropathy after skull base surgery. However, addressing the above suggestions would enhance the paper’s clarity and contribute to its scientific rigor.
Author Response
We are very grateful to the reviewers for their insightful comments and suggestions, which would undoubtedly help us to improve our manuscript immensely. As indicated in the responses below, we have taken all their comments and suggestions into account when generating the revised version of the manuscript. Responses to the reviewers’ comments appear after the arrows, in blue text.
Reviewer 4:
The paper titled “Clinical effectiveness of mirogabalin besylate for trigeminal neuropathy after skull base surgery. Illustrative cases.” discusses the use of mirogabalin besylate, a selective ligand for the α2δ subunit of voltage-gated calcium channels, for postoperative trigeminal neuropathy after skull base surgery. The authors present two illustrative cases of patients who experienced tingling and numbness in the face after their respective surgeries, and they administered mirogabalin orally to manage the symptoms. In both cases, the patients showed improvement in their facial numbness, indicating that mirogabalin may be effective in providing pain relief for trigeminal neuropathy after skull base surgery.
→
Thank you very much for your review.
Suggestions for Improvement:
1.Introduction : The introduction could benefit from providing a more comprehensive overview of the existing literature general pain management for trigeminal pathy including chronic conditions and inventional minimally invasive methods. Known adverse effects of mirogabalin should be stated.
→
Thank you very much for your comments.
As the reviewer indicated, we have described the general guideline of trigeminal neuropathy.
Side effect of mirogabalin besylate was described in the section of Discussion.
For trigeminal neuropathy, carbamazepine and oxcarbazepine are drugs of first choice. Lamotrigine, gabapentin, pregabalin, botulinum toxin type A and baclofen are used either alone or as add-on therapy. Surgery should be considered if the pain is poorly controlled. Trigeminal microvascular decompression is the first-line surgery in patients with trigeminal neurovascular conflict [Chong MS. Cleve Clin J Med. 2023; Pergolizzi JV Jr. Expert Opin Pharmacother. 2022].
Chong MS, Bahra A, Zakrzewska JM. Guidelines for the management of trigeminal neuralgia. Cleve Clin J Med. 2023 Jun 1;90(6):355-362.
Pergolizzi JV Jr, Gharibo C, Magnusson P, Breve F, LeQuang JA, Varrassi G. Pharmacotherapeutic management of trigeminal neuropathic pain: an update. Expert Opin Pharmacother. 2022 Jul;23(10):1155-1164.
2.Methodology: A clear description of the assessment tools used to measure pain relief would also be helpful.
→
Thank you very much for your comments. As the reviewer indicated, we have added a clear description of the assessment tools.
The VRS is a simple tool limited to a few statements, and it appears to be the most usable tool for pain assessment in cognitively impaired subjects.
The VRS is a four-point scale and consists of a list of adjectives describing various levels of pain intensity (0= no pain, 1= mild pain, 2= moderate pain, and 3= severe pain) [Jensen MP. Pain. 1986; Haefeli M. Eur Spine J. 2006].
Jensen MP, Karoly P, Braver S. The measurement of clinical pain intensity: a comparison of six methods. Pain. 1986;27:117-26.
Haefeli M, Elfering A. Pain assessment. Eur Spine J. 2006;15:S17-24.
3.Discussion: Postsurgical trigeminal neuralgia is a rare condition. What about the usage of miragabalin for chronic trigeminal neuralgia? Are there any previous studies reporting on that? The discussion section should be expanded to interpret the results and compare them with previous studies.
→
Thank you very much for your review. We have discussed about the possibility of mirogabalin for the patients with chronic trigeminal neuralgia in the revised manuscript.
Recent case reports demonstrated that mirogabalin is effective for the patients with chronic trigeminal neuralgia, such as trigeminal trophic syndrome [Matsuda KM. Eur J Dermatol. 2020; Tang H. Mediators Inflamm. 2023; Kim JY. Korean J Pain. 2021; Noma N. J Indian Prosthodont Soc. 2021; Kikuchi K .Invest Ophthalmol Vis Sci. 2023]. Mirogabalin may be efficiently used for the patients with general trigeminal neuralgia in the chronic stage.
Matsuda KM, Tanaka-Mizutsugu H, Kishi Y, Hino H, Kagami S. A case of trigeminal trophic syndrome responding to mirogabalin. Eur J Dermatol. 2020:doi: 10.1684/ejd.2020.3746.
Tang H, Lu J, Duan Y, Li D. The Clinical Application and Progress of Mirogabalin on Neuropathic Pain as a Novel Selective Gabapentinoids. Mediators Inflamm. 2023;2023:4893436.
Kim JY, Abdi S, Huh B, Kim KH. Mirogabalin: could it be the next generation gabapentin or pregabalin?Korean J Pain. 2021;34:4-18.
Noma N, Ozasa K, Young A. Altered somatosensory processing in secondary trigeminal neuralgia: A case report.J Indian Prosthodont Soc. 2021;21:308-310.
Kikuchi K, Tagawa Y, Murata M, Ishida S. Effects of Mirogabalin on Hyperalgesia and Chronic Ocular Pain in Tear-Deficient Dry-Eye Rats.Invest Ophthalmol Vis Sci. 2023;64:27.
4.Please, discuss limitations of use and side effects of the drug.
→
Thank you very much for your comments. We have added the limitations and side effects of this drug.
The most frequent adverse drug reactions including nasopharyngitis, somnolence, dizziness, peripheral edema, drowsiness and cerebellar ataxia were not observed. However, careful dosing and titration may be necessary. In particular, dizziness and somnolence are careful side effects, occurring in a certain proportion, which is why it is recommended to refrain from taking the medicine before driving. For individuals leading a daily life that necessitates driving, there is a potential challenge in timing the administration to achieve the desired analgesic effect. Furthermore, in the case of elderly individuals, it demands attention due to the increased risk of falls. Additionally, the medication is comparatively more expensive than typical anti-inflammatory analgesics, posing an economic burden [Finnerup NB, Lancet Neurol. 2015].
Finnerup NB, Attal N, Haroutounian S, McNicol E, Baron R, Dworkin RH, Gilron I, Haanpää M, Hansson P, Jensen TS, Kamerman PR, Lund K, Moore A, Raja SN, Rice AS, Rowbotham M, Sena E, Siddall P, Smith BH, Wallace M. Pharmacotherapy for neuropathic pain in adults: A systematic review and meta-analysis and updated NeuPSIG recommendations. Lancet Neurol 2015;142:162–73.
Limitation
Herein we report the usage of mirogabalin in patients that suffer mild trigeminal neuropathy. As both patients are aware of taking medication for neuropathic pain, there is a possibility of the placebo effect intervening. Therefore, we must discuss confounding factors that might have influenced the observed outcomes. Mirogabalin is expected to be an important new treatment option for the patients with trigeminal neuropathy after skull base surgery, which should be followed by a study that includes patients with se-vere neuropathy. Admittingly, it might only be that mirogabalin allowed some/minor relief to these two patients to give some help to accelerate the course. In general, NSAIDs and acetaminophen are effective for wound pain but not for neuropathic pain. Therefore, the effect of mirogabalin should be compared with other antineuropathic drugs in-cluding pregabalin. In particular, pregabalin is a common prescription drug, which is typically used to treat neuropathic pain, anxiety, and restless leg syndrome [18]. Detailed information (more strict follow up, formal scales of pain/life quality, etc.) is needed to more clearly understand the effect of mirogabalin. Further studies are required to de-termine appropriate dose adjustments on patient weight and body surface area. Fur-thermore, most of the studies were conducted with only Asian patients, prompting the question if the results are generalizable to the general population. Long-term evalua-tions of safety and efficacy in other races are still needed. The level of evidence of this case report is poor. The VRS is a simple tool limited to a few statements, and it appears to be the most usable tool for pain assessment in cognitively impaired subjects. However, other pain rating scales, including Visual Analogue Scale and Numerical/numeric Rating Scale are needed to evaluate detailed clinical course [Williamson A. J Clin Nurs. 2005]. A prospective study with large number of patients is needed to confirm the efficacy and safety of mirogabalin to treat trigeminal neuropathy after skull base surgery.
Williamson A, Hoggart B. Pain: a review of three commonly used pain rating scales. J Clin Nurs. 2005:14:798-804.
5.It would be beneficial to discuss confounding factors that might have influenced the observed outcomes.
→
Thank you very much for your comments. We have discussed confounding factors in the revised manuscript.
As both patients are aware of taking medication for neuropathic pain, there is a possibility of the placebo effect intervening. Therefore, we must discuss confounding factors that might have influenced the observed outcomes.
6.Conclusion: The conclusion, especially this in the abstract, is too brief; consider adding a more comprehensive summary of the study’s findings and emphasize the potential implications and significance of the results for clinical practice.
→
Thank you very much for your comments. We have modified the conclusions in the revised manuscript.
Abstract
Mirogabalin may show significant pain relief for the patients with trigeminal neuropathy after the skull base surgery. Further studies using a larger number of patients are warranted to confirm these findings.
Main text
Mirogabalin may show significant pain relief for the patients with trigeminal neuropathy after the skull base surgery. Mirogabalin may be expected to be an important new treatment option for the patients after skull base surgery. Furthermore, it may be also helpful for not only postoperative treatment but also preoperative management for the tumors involving trigeminal nerve, such as schwannoma and meningioma. However, drug efficacy studies should be carried out by comparison with the effect of placebo or other painkillers. Further studies using a larger number of patients are warranted to confirm these findings.
- Limitations: Given that this is a case report with a limited sample size, the paper should caution readers about the generalization of the findings. This caution including a statement on the need for further research with larger cohorts should be indicated as a short "Limitations" section in the paper.
→
Thank you very much for your review. As the reviewer indicated, we have added the limitation section, described above.
8.Overall, the study presents promising results regarding the effectiveness of mirogabalin besylate for trigeminal neuropathy after skull base surgery. However, addressing the above suggestions would enhance the paper’s clarity and contribute to its scientific rigor.
→
Thank you very much for your nice comments. We have modified the manuscript as the reviewer indicated.

Round 2
Reviewer 1 Report
The authors addressed the comments. In my opinion, the manuscript can be published.